# Factors that Decrease Sedentary Behavior in Community-Dwelling Elderly People: A Longitudinal Study

**DOI:** 10.3390/medicina56040157

**Published:** 2020-04-01

**Authors:** Yutaka Owari

**Affiliations:** Department of Judo Therapy, Shikoku Medical College Utadu, Kagawa 769-0205, Japan; seikotsuin@nifty.com; Tel.: +81-87-41-2320

**Keywords:** health class, sedentary behavior, elderly people, longitudinal study

## Abstract

*Background and objectives:* This study was to clarify how the frequency of participation in a health class affected the reduction in sedentary behavior after two years, and whether decreases in sedentary behavior in elderly people who participated in a health class persisted two years after the end of an intervention. *Materials and Methods:* This study was longitudinal, and the results of a previous study conducted in 2017 were added to the findings of a different year. The participants were elderly health class members at a community dwelling in Japan who participated between 2016 and 2018. Of the 86 participants that were enrolled, the data of 80 were collected in 2016. A year later, in a 2017 follow-up, the number of participants was 80, and two years later, in a 2018 follow-up, there were 71 participants. *Results:* There was a significant difference with regards to the reduction of the sedentary behavior rate between two different groups (a health class participation rate of more than 75% and the other with less than 75%) two years later. However, no difference in sedentary behavior rate was found between the two groups (intervention group and control group) at a two-year follow-up, despite observing differences after one year. *Conclusions:* Continuous participation in a health class may help reduce sedentary behavior. After two years, “Active Guide” brochures and documents may not help in reducing sedentary behavior, despite seeing improvements after one year. Persistent social participation may have a more lasting effect than one-off interventions in reducing the sedentary behavior ratio.

## 1. Introduction

Of the world’s major countries by population, Japan has the highest rate of sedentary behavior [1]. The average sedentary behavior rate in the major countries was 300 min/d, in which Portugal and Japan had the lowest (150 min/d) and highest (420 min/d), respectively [1]. These values are based on participants’ self-reports, and are less accurate, but general trends can be seen. Compared with other countries, Japan has longer sitting times and sedentary behavior rates increase with age [2]. Based on the National Health and Nutritional Examination Survey conducted in the United States, sedentary behavior in those in their 60 s increased by 60 min, and in those in their 70 s increased by more than 120 min, compared to in those in their 30s [3]. Here, the term “sedentary behavior” does not refer to inactivity (i.e., lack of medium to high intensity physical activity recommended in the physical activity policy), which is used in research for promoting physical activity [4]. Instead, it refers to all wakeful behaviors with energy consumption of 1.5 METs or less, which include sitting and lying positions [5].

Furthermore, increases in sedentary behavior lead to increases in health risks. There are several reports that clarify the relationship between sedentary behavior and various health risks, such as overall mortality, cardiovascular disease, cancer, and diabetes [6,7,8,9]. Van der Ploeg et al. [10] showed that the total mortality risk for those who sat for more than 11 h per day was 40% higher than for those who sat for less than 4 h, which was more pronounced in the elderly. Furthermore, long-term sedentary behavior was found to be associated with a decline in daily functions in the elderly, such as motor and cognitive function [6,11]. Siddarth et al. [12] showed that prolonged sedentary behavior reduced cognitive ability and the area of the brain involved in memory formation. Based on the Statistics Bureau of the Ministry of Internal Affairs and Communications in Japan, the elderly population over 65 in Japan was 35.57 million as of September 15, 2018, which accounts for 28.4% of the total population. The National Institute of Population and Social Security Research estimated that Japan’s elderly population was 38.8% in 2050 [13]. Therefore, efforts to reduce sedentary behavior in elderly people are of national importance.

There are also worldwide efforts to reduce sedentary behavior [14,15,16]. In Japan, interventions were developed to address various sedentary behaviors that affect both the physical and mental health of elderly people [17]. In a previous study, we showed in a one-year RCT study that intervention with an “Active Guide” brochure [18] was effective in reducing sedentary behavior in elderly people who participated in health classes [19]. However, few studies have reported on the long-term effects of intervention.

Therefore, in this study, we clarified how the frequency of participation in a health class affected the reduction in sedentary behavior after two years, and whether decreases in sedentary behavior in elderly people who participated in a health class persisted two years after the end of an intervention.

## 2. Materials and Methods

### 2.1. Study Design

This study was longitudinal, and the results of a previous study conducted in 2017 [19] were added to the findings of a different year. The participants in this study were the same as in the previous study. The following two hypotheses were proposed.

**Hypothesis** **1.**
*The effects of reducing sedentary behavior are dependent on participation in health classes.*


**Hypothesis** **2.**
*The sustained effects post intervention on reducing sedentary behavior are limited.*


### 2.2. Participants and Data Collection

The participants were college health club members in Utadu-cho, Kagawa Prefecture, Japan (population of approximately 18,450) that participated in the study between 2016 and 2018. Of the 112 participants in the health class, 90 consented to an initial survey. Twenty-two participants were excluded, because they did not participate in more than 75% of the 12 sessions during the trial period. The study was conducted between 20 July and 10 September 2016 [19]. The data of 4 participants who canceled their enrollment from this study were excluded from the analysis. Therefore, the results from the remaining 86 participants were used as baseline. All participants were 65 years or older. A similar follow-up study was conducted between 20 July and 15 September 2017. Of the initial survey respondents and excluding 6 that could not be surveyed, the data from 80 participants were analyzed (71.9 ± 5.5 years) in a previous report [19]. Survey two years later, 74 participants consented to a follow-up in 2018 (between April 29 and May 31) and were requested to fill out a questionnaire by mail and be attached to an activity meter. Of these participants, the data from 71 were eligible for analysis (22 men; 49 women; average age, 73.9 ± 5.5 years).

### 2.3. Assessment of Factors

#### 2.3.1. Basic Attributes

Various socio-demographic factors and physical factors were assessed by a self-administered questionnaire. Subjects were also evaluated on the following parameters in 2016 and 2017: age (years) and Body Mass Index (BMI, kg/m^2^) [19]. The obtained data were added to a 2018 survey.

#### 2.3.2. Health Class

In a previous study (2016–2017), 86 of 90 collaborators participated in more than 75% of 12 sessions during the trial period. In the 2017–2018 study, the number of data obtained was reduced to 71. The breakdown of 71 people was 30 (more than 75% attended health class) and 41 (less than 75% attended health class). Health class sessions were held once a month for 90 min, and mainly consisted of lectures (45 min) and exercises (45 min) for health promotion. From 2016–2017, the average number of participants per event was 50.2 ± 10.3 (minimum, 29; maximum, 62). During 2017–2018, the average number of participants per event was 36.5 ± 8.7 (minimum, 20; maximum, 50). Participants in a health class are to attend each month for two years in principle, but this is not mandatory. The number of registrants was 121 at the end of 2018. The lectures encompassed 27 topics: 14 focused on muscle, bone, and joints, 6 on medical diseases, 7 on health care, from 2016–2017. Additionally, the same lectures were repeated from 2017–2018. Lectures and exercises were facilitated by doctors, physical therapists, occupational therapists, nurses, acupuncturists, judo therapists, and athletic trainers.

#### 2.3.3. Physical Activity

In this study, physical activity was assessed using a 3-axis accelerometer (Active Style Pro HJA-750C, OMRON Healthcare, Japan) for 10 consecutive days, as reported in a previous study [20]. Taking activity measurements from the Active Style Pro has been previously validated [21,22,23]. Participants were requested to wear the accelerometer on their waist at all times, except during certain activities, such as swimming and bathing. The criterion for accepting accelerometer data was when the acceleration signal zero continued for 60 min or more. Participants who wore the accelerometers for more than 10 h per day, for more than 7 d, were included in the analysis. The standard deviation of the data for 10 s was defined as the average acceleration. The units for the following parameters are as follows: Exercise: Σ, [metabolic equivalents × hours per week (METs h/week)], Number of steps (steps/d), Walking time (min/d), and physical activity (≤1.5METs, 1.6–2.9 METs, 3–5.9 METs) (min/d). Since the mean physical activity did not show a normal distribution (Shapiro–Wilk test: *p* < 0.05), the physical activity of each participant was assigned a median of 7 d. We adopted the median of each participant to calculate the average in each group. Physical activity and sedentary behavior were assessed at baseline, as well as one and two years later.

#### 2.3.4. Active Guide

The Ministry of Health, Labor and Welfare aims to prevent individual lifestyle-related diseases, as well as severe lifestyle-related diseases, as part of efforts of the Second National Health Promotion Movement (Health Japan 21 (Secondary)). In 2013, the “Physical Activity Guidelines for Health and Health Active Guide” was developed to promote healthy lifestyles, in order to increase life expectancy and improve the quality of the social environment. The “Active Guide”, based on the “Physical Activity Reference 2013”, introduced “Plus 10” as a catchphrase to promote daily physical activity. “Plus 10” proposes that one moves one’s body for 10 min more than the current activity level, to maintain a healthy lifestyle and increase life expectancy (Miyachi, Tripette, Kawakami, and Murakami, 2015). In addition, the “Active Guide” recommends standing, walking, or performing daily chores for people over the age of 65.

#### 2.3.5. Hypothesis 1 Methods

In addition to the interventions proposed by the Active Guide, participation in a health class was considered as one of the factors that decreased sitting time. Therefore, we were determined to clarify the effect of reducing sitting time by participation in a health class. Between 2016 and 2017, all participants attended over 75% of the health classes. However, due to changes in attendance rates after 2017, the participants were divided into two groups: those who attended 75% or more of the health class (Group A), and those who attended less than 75% (Group B). We adopted a similar classification criterion in Hypothesis 1, because the initial eligibility criterion was a participation rate of over 75% in health classes. Additionally, participants who received the “Active Guide” brochure and related documents outlining the benefits of reducing sedentary behavior were 14 (46.7%) in Group A, and 21 (51.2%) in Group B.

#### 2.3.6. Hypothesis 2 Methods

In a previous study [19], we showed that intervention may reduce sedentary behavior in elderly people one year later. However, additional studies were required to assess whether the decline in sedentary behavior persisted after two years. The second phase study involved a similar follow-up survey and was conducted between 29 April and 31 May 2018.

#### 2.3.7. Previous Study Methods

The RCT study consisted of two groups of elderly people living in Japan. Our RCT study, however, instead of random sampling, used random assignment. Shortly after receiving their baseline results, the intervention group also received the “Active Guide” brochure and related documents outlining the benefits of reducing sedentary behavior. Members of the control group received only the baseline test results.

The sample size was calculated using significance level, power, difference between two groups, and standard deviation (SD) by paired t-test. To have a sample exceeding the calculated sample size of 39 per group is good, and could be explained in those terms, rather than as a deliberate sampling choice to have 10% attrition allowance.

### 2.4. Statistical Analyses

The data analysis was conducted by a researcher blinded to the groupings of the participants. Continuous variables were presented as the mean ± SD. Welch’s t-test was used to compare the averages of continuous variables. If there were variables present that could affect the results, they were adjusted using a multiple regression analysis by these variables, prior to intervention. The threshold for significance was *p* < 0.05. All calculations were performed using STATA, version 14 (STATA Corp LLC., 4905 Lake way College Station, Texas 77845-4512 USA).

### 2.5. Ethical Consideration

This study was approved by the Shikoku Medical University Ethics Screening Committee (approval number: H27-3: May 13, 2016; H27-7: May 12, 2017; H28-6: April 06, 2018), and written informed consent was obtained from each participant. This study was registered as a randomized, controlled trial by the University Hospital Medical Information Network (UMIN) (registration number UMIN000027781).

## 3. Results

### 3.1. Hypothesis 1 Results

There were a total of 86 participants at baseline; 3 were lost to follow-up, 2 were hospitalized, and 1 was rejected. Therefore, 80 data were used (Figure 1, previous study). As shown in Table 1, the two groups were homogeneous, except for the BMI in the randomized assignment and baseline. Among the 80 subjects, 3 were lost to follow-up, 2 were certified for long-term care, 2 lacked measurements, 1 had an incomplete questionnaire, and 1 was hospitalized in 2018. Participants in 2018 regrouped into two groups, based on health class participation: Group A, participation rates of greater than 75% (number of participants, 30; average age, 74.2 ± 5.9 years), and Group B, less than 75% (number of participants, 41; average age, 73.7 ± 5.2 years) (Figure 1, Hypothesis 1). There was a significant difference compared to the previous year in sedentary behavior (≤1.5 METs); reduction between Groups A (−6.7%) and B (−2.4%) (*p* = 0.041) (Table 2).

### 3.2. Hypothesis 2 Results

Hypothesis 2′s participants (71 participants), the intervention (number of participants, 35: average age, 73.5 ± 5.7 years) and control group (number of participants, 36: average age, 74.3 ± 5.3 years) (Figure 1, Hypothesis 2), were the same as in Hypothesis 1. No differences in sedentary behavior rate were found between groups (intervention group, 50.7%/d, control group. 50.6%/d; *p* = 0.872) at the two-year follow-up (Table 3), despite observing differences after one year (Table 4). On the contrary, the parameters before and after the two-year follow-up were significantly lower in the control group than in the intervention group: ≤1.5METs (%/) (−2.1 vs. −7.0; *p* = 0.034), 1.6~2.9 METs (%/d) (6.4 vs. 10.1; *p* = 0.041), 3~5.9 METs (%/d) (−1.5 vs. 2.1; *p* = 0.044) (Table 3).

### 3.3. Previous Results

At baseline, the sedentary behavior rates were 54.9% and 55.2%/d in the intervention and control group, respectively (*p* = 0.856). As shown in Table 1, there was no difference in sedentary behavior. During the follow-up period, the differences in sedentary behavior rate (before and after intervention, %) were −2.2% and 2.5%/d in the intervention and control groups, respectively (*p* = 0.007). There was a significant difference in sedentary behavior (%) between the two groups during the follow-up period. Differences in BMI during the follow-up were also observed. The sedentary behavior rates were 52.7% and 57.7%/d in the intervention and control groups, respectively, (*p* = 0.033) at follow-up (Table 4). Even after adjusting for BMI by multiple regression analysis, a change in sedentary behavior (%) between the two groups was observed.

## 4. Discussion

There was a statistically significant difference in the rate of reduction in physical activity (≤1.5 METs) between the group that took the health class and the group that did not. On the other hand, there was no significant difference between the intervention group and the control group. The “Active Guide” brochure and additional information handouts were effective one year after the intervention; they might not be valid after two years in community-dwelling Japanese elderly people.

### 4.1. Hypothesis 1

There was a significant difference in sedentary behavior (%/d) between the group with health class attendance rates of 75% or more (Group A) and the group with attendance rates of less than 75% (Group B). This discrepancy may be due to the subjects in Group A reducing their sedentary behavior by attending monthly health classes. Group A participated in health classes (a type of social participation) for at least two years, which may increase their awareness of improving health, thereby reducing their sedentary behavior. Although social participation has been shown to be useful for maintaining and improving the health of the elderly, the relationship with sedentary behavior is not fully understood. Kikuchi et al. [24] showed that social participation for elderly people was useful in reducing sedentary behavior.

The reduction in sedentary behavior (≤1.5 METs) was as large as −6.7% for Group A, and −2.4% for Group B. This may be due to the fact that there were nine dropouts in 2018, compared to the previous year, but the average percentage of those nine who resigned in 2017 was 64.7% (≤1.5 METs).

There are some limitations to this study. First, instead of random sampling, random assignment was used. We focused on elderly people who participated in a health club held at vocational schools in small cities. Since the subjects were already members of a health club, they were more likely to be health-oriented than the average elderly person. Their baseline sedentary behavior was already significantly lower (56.4–53.8%) than the average Japanese elderly (60.2%, 73.1 years) (Honda et al. [25]). Since this group was relatively active, there may be little impact of this intervention in the population studied. The average elderly Japanese person may benefit more from this intervention than the study group. Second, although a simple random assignment was performed, there was a significant difference in BMI between the intervention and control groups.

In future studies, in addition to clarifying the sitting rate, the effect of its interruption should be assessed, in order to determine methods to improve the health of the elderly.

### 4.2. Hypothesis 2

Two years after intervention, there were no changes in sedentary behavior rate (%) between the two groups, as well as no differences in sedentary behavior (%) between the two groups. Currently, there are no randomized controlled trials that clarify the long-term effects (2 years) of interventions on reducing sedentary behavior in the elderly. In a review of a large-scale study of cardiovascular disease prevention, Berkman et al. [26] showed that the combination of 36 studies resulted in a 0.65% reduction in cardiovascular disease risk. Of seven studies, none showed decreases in mortality rates. Our intervention was also less effective for the elderly after a prolonged period.

However, changes in sedentary behavior (%/d) of the control group were significantly lower than that of the intervention group. Furthermore, changes in other parameters (1.6~2.9 METs (%/d), 3~5.9 METs (%/d)) of the control group were significantly higher than that of the intervention group. This discrepancy may have been due to the effects on the parameters of exercise intensity of the participants that were excluded in the 2018 study (5 and 4 from the intervention and control group, respectively). In 2017, the average percentages of sedentary behavior of the excluded participants in the intervention and control groups were 60.0% and 70.8%/d. The dropout of these participants in 2018 may have led to a decrease in the control group’s sitting position ratio in 2018. Conversely, the average percentages of 1.6 to 2.9 METs and 3 to 5.9 METs of the excluded participants in the intervention group in 2017 were 33.4 and 6.9%/day, respectively, whereas the average percentages of 1.6 to 2.9 METs and 3 to 5.9 METs in the control group were 25.2 and 4.5%/day, respectively. Excluding these participants in the 2018 study may have increased in the ratio of 1.6 to 2.9 METs and 3 to 5.9 METs in the control group in 2018.

### 4.3. Previous Study 3

There was a significant difference in sedentary behavior (%) between the two groups. Changes in sedentary behavior were −2.2% and +2.5% in the intervention and control groups, respectively. We found that the “Active Guide” brochure and additional related documents reduced sedentary behavior for one year.

## 5. Conclusions

In conclusion, (1) continuous participation in health clubs may help reduce sedentary behavior. (2) After two years, the use of the “Active Guide” brochure and related documents may not reduce the sedentary behavior in elderly people living in Japan, despite observing a reduction after one year. Persistent social participation may have a more lasting effect than one-off interventions in reducing the sedentary behavior ratio.

## Figures and Tables

**Figure 1 medicina-56-00157-f001:**
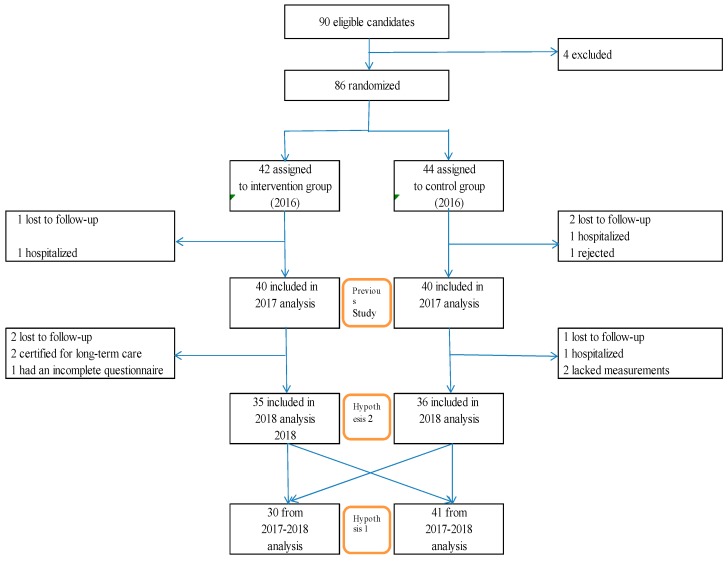
Randomization and follow-up.

**Table 1 medicina-56-00157-t001:** Characteristics of enrolled subjects in 2016.

	Intervention Group					Control Group					
	Mean	±	SD	Minimum	Maximum	Mean	±	SD	Minimum	Maximum	*P*
Number of subjects	40 (12 men, 28 women)					40 (12 men, 28 women)					
Age (years)	72.6	±	5.5	65	85	71.1	±	5.5	65	85	0.197
BMI (kg/m^2^)	21.8	±	2.6	14.9	26.7	23.5	±	2.7	18.7	29.1	**<0.001**
Exercise (METs h/w)	5.1	±	2.2	0.4	9.7	5.2	±	2.2	1.6	9.7	0.800
Number of steps (steps/d)	5728.2	±	2818	569.9	12,230.1	5746.5	±	2339	1585.4	10,915.9	0.975
Walking time (minutes/d)	83.2	±	38.3	20.9	177.7	87.4	±	26.9	31.3	146.7	0.572
≤1.5 METs (%/d)	54.9	±	9.7	35.4	75.5	55.2	±	10.2	38.3	79.9	0.856
1.6~2.9 METs (%/d)	35.5	±	7.0	19.4	49.0	34.9	±	8.5	16.9	52.3	0.719
3~5.9 METs (%/d)	8.9	±	3.9	2.8	17.0	9.2	±	3.5	0.8	15.6	0.831

BMI: body mass index (kg/m^2^); METs: metabolic equivalents. Values in bold are significant (*p* < 0.05).

**Table 2 medicina-56-00157-t002:** Characteristics of enrolled subjects in 2018: Groups A and B.

	Group A					Group B					
	Mean	±	SD	Minimum	Maximum	Mean	±	SD	Minimum	Maximum	*P*
Number of subjects	30 (8 men, 22 women)					41 (14 men, 27 women)					
Age (years)	74.2	±	5.9	67	87	73.7	±	5.2	67	87	0.696
BMI (kg/m^2^)	22.6	±	2.7	17.4	26.4	22.7	±	2.9	14.9	26.8	0.878
Exercise (METs h/w)	4.8	±	2.2	1.6	9.7	5.9	±	2.9	1.3	9.6	0.078
Number of steps (steps/d)	5751.6	±	2639.2	1374.9	11,291.0	6460.0	±	3001.9	985.4	13,973.3	0.305
Walking time (min/d)	79.7	±	31.5	38.6	167.4	92.6	±	41.2	20.9	177.7	0.157
≤1.5 METs (%/d)	47.9	±	10.0	30.2	66.9	52.7	±	14.2	23.8	83.5	0.160
Changes in ≤1.5 METs (%/d)	−6.7	±	8.0	−23.6	11.2	−2.4	±	7.0	−23.4	13.6	**0.041**
1.6~2.9 METs (%/d)	35.5	±	6.9	18.9	14.7	34.4	±	8.4	13.1	28.7	0.373
Changes in 1.6~2.9 METs (%/d)	6.4	±	5.8	−9.3	14.7	5.3	±	7.7	−10.9	28.7	0.486
3~5.9 METs (%/d)	8.4	±	3.5	2.8	15.2	8.9	±	4.7	2.9	22.5	0.622
Changes in 3~5.9 METs (%/d)	2.0	±	4.9	−1.8	1.3	3.4	±	3.0	−1.6	1.2	0.147

BMI: body mass index (kg/m^2^); METs: metabolic equivalents. Values in bold are significant (*p* < 0.05).

**Table 3 medicina-56-00157-t003:** Characteristics of enrolled subjects in 2018: a two-year follow-up.

	Intervention Group					Control Group					
	Mean	±	SD	Minimum	Maximum	Mean	±	SD	Minimum	Maximum	*P*
Number of subjects	35 (12 men, 23 women)					36 (10 men, 26 women)					
Age (years)	73.5	±	5.7	67	87	74.3	±	5.3	67	87	0.613
BMI (kg/m^2^)	22.7	±	2.7	14.9	26.8	22.6	±	2.5	14.9	26.6	0.980
Exercise (METs h/w)	5.3	±	2.3	1.6	9.7	5.2	±	2.2	1.8	9.7	0.886
Number of steps (steps/d)	5728.1	±	2395.4	1585.4	10,915.9	5916.0	±	2824.6	923.7	12,230.1	0.764
Walking time (min/d)	88.3	±	27.9	31.3	146.7	85.6	±	38.9	20.9	177.7	0.739
≤1.5 METs (%/d)	50.7	±	12.8	30.2	73.1	50.6	±	14.4	23.8	83.5	0.872
Changes in ≤1.5 METs (%/d)	−2.1	±	8.2	−23.6	11.2	−7.0	±	10.9	−50.4	13.6	**0.034**
1.6~2.9 METs (%/d)	42.8	±	12.4	22.9	18.4	43.8	±	11.6	13.1	28.7	0.117
Changes in 1.6~2.9 METs (%/d)	6.4	±	7.7	−3.4	14.7	10.1	±	9.7	−10.9	22.5	**0.041**
3~5.9 METs (%/d)	8.8	±	4.1	2.8	17.0	10.6	±	4.5	4.1	22.5	0.094
Changes in 3~5.9 METs (%/d)	−1.5	±	2.2	−0.4	0.2	2.1	±	2.6	−1.8	1.3	**0.044**

BMI: body mass index (kg/m^2^); METs: metabolic equivalents. Values in bold are significant (*p* < 0.05).

**Table 4 medicina-56-00157-t004:** Characteristics of enrolled subjects in 2017: a one-year follow-up.

	Intervention Group					Control Group					
	Mean	±	SD	Minimum	Maximum	Mean	±	SD	Minimum	Maximum	*P*
Number of subjects	40 (12 men, 28 women)					40 (12 men, 28 women)					
Age (years)											
BMI (kg/m^2^)	21.9	±	2.7	14.9	26.7	23.3	±	2.6	19.1	29.1	**<0.001**
Exercise (METs h/w)	5.5	±	2.8	0.5	10.9	5.0	±	2.5	0.9	10.7	0.340
Number of steps (steps/d)	6138.5	±	3188	845.1	13,973.3	5784.3	±	2503	985.4	11,291.0	0.582
Walking time (minutes/d)	87.3	±	42.8	32.7	192.1	84.3	±	31.5	38.6	167.4	0.717
≤1.5 METs (%/d)	52.7	±	10.9	35.4	75.5	57.7	±	9.8	36.5	74.3	**0.033**
Changes in ≤1.5 METs (%/d)	−2.2	±	5.9	16.6	8.0	2.5	±	8.8	−8.6	23.3	**0.007**
1.6~2.9 METs (%/d)	36.4	±	8.0	19.4	49.0	33.7	±	7.4	19.6	47.4	0.117
Changes in 1.6~2.9 METs (%/d)	0.9	±	4.1	2.6	−1.9	−1.2	±	6.3	−2.4	1.1	0.098
3~5.9 METs (%/d)	10.3	±	4.6	2.8	17.0	8.5	±	4.0	1.9	15.8	0.067
Changes in 3~5.9 METs (%/d)	1.4	±	2.2	−0.4	0.2	−0.7	±	1.9	−1.8	1.3	0.061

BMI: body mass index (kg/m^2^); METs: metabolic equivalents. Values in bold are significant (*p* < 0.05).

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
