# Peer review of "Factors that Decrease Sedentary Behavior in Community-Dwelling Elderly People: A Longitudinal Study"

_medicina, 2020, doi:10.3390/medicina56040157_

Round 1

Reviewer 1 Report

The authors have presented an interesting study on the decrease in sedentary behaviors. However, the manuscript need some revise.

Abstract section: It is not clear what the study is, it is commented that it is a longitudinal study and later reference is made to data from a randomized clinical trial. the reviewer advises the authors to avoid reader confusion in this regard. In results, the reviewer advises the authors to make a brief description of the sample profile. the conclusions are a bit poor for the ambition of the study or the studies carried out.

Introduction section the reviewer congratulates the authors for the conciseness of this section and the selection of bibliographic references.

Materials and Methods section. Study design: The reviewer suggests that the authors improve the description of the studies carried out. The reviewer also suggests that the authors move the objectives to the end of the introduction. Subjects and data collection: The reviewer advises authors to include initial eligibility criteria.

Point 2.3.6 the authors cannot refer to the results of a study in the materials and methods section.

Point 2.3.8 The reviewer suggests that the authors describe the statistical analysis in more detail.

Results section. the reviewer advises the authors to transfer figure 2 to the methods section. Likewise, the reviewer advises the authors to write the results so that the found results are more clearly expressed. the use of tables should reduce the text to punts of greater emphasis.

Discussion section: The reviewer suggests that the authors initially highlight the main findings linked to the main objectives and develop them in the discussion, without the need to present them following the chronological sequence of the studies carried out. Conclusions Again the reviewer suggests to the authors to clarify what the contributions of the study  are.

Author Response

Reviewer 1

The authors have presented an interesting study on the decrease in sedentary behaviors. However, the manuscript need some revise.

Abstract section: It is not clear what the study is, it is commented that it is a longitudinal study and later reference is made to data from a randomized clinical trial. the reviewer advises the authors to avoid reader confusion in this regard. In results, the reviewer advises the authors to make a brief description of the sample profile. the conclusions are a bit poor for the ambition of the study or the studies carried out.

(Answer)

I rewrote it as follows.

Abstract: Background and objectives: This study was to clarify how the frequency of participation in a health class affect reduction in sedentary behavior after two years and whether decreases in sedentary behavior in elderly who participated in a health class persist two years after the end of intervention. Materials and Methods: This study was longitudinal, and the results of a previous study conducted in 2017 were added to the findings of a different year. The participants were elderly health class members at a community dwelling in Japan who participated between 2016 and 2018. Of the 86 participants that were enrolled, the data of 80 were collected in 2016. A year later, in a 2017 follow-up, the number of participants was 80, and two years later, in a 2018 follow-up, there were 71. Results: There was a significant difference from the previous year in sedentary behavior rate of reduction between groups with health class participation rate of greater than 75% and less than 75%. Furthermore, no difference was found between groups in sedentary behavior rate at a two-year follow-up compared to the previous year despite observing differences after one year. Conclusions: Continuous participation in a health class may help reduce sedentary behavior. After two years, “Active Guide” brochures and documents may not help in reducing sedentary behavior despite seeing improvements after one year. Persistent social participation may have a more lasting effect than one-off interventions in reducing the sedentary behavior ratio.

Introduction section the reviewer congratulates the authors for the conciseness of this section and the selection of bibliographic references.

(Answer)

Thank you for your review.

Materials and Methods section. Study design: The reviewer suggests that the authors improve the description of the studies carried out. The reviewer also suggests that the authors move the objectives to the end of the introduction. Subjects and data collection: The reviewer advises authors to include initial eligibility criteria.

(Answer)

1.(2.3.5-2.3.7): L.129-154.

(I have improved the description.)

2.3.5. Hypothesis 1 methods

In addition to the interventions proposed by the Active Guide, participation in health classes was considered as one of the factors that decreased sitting time. Therefore, we determined to clarify the effect of reducing sitting time by participation in health classes. Between 2016 and 2017, all participants attended over 75% of the health classes. However, due to changes in attendance rates after 2017, the participants were divided into two groups: those who attended 75% or more of the health classes (Group A) and those who attended less than 75% (Group B). We adopted a similar classification criterion in Hypothesis because the initial eligibility criterion was a participation rate of over 75% in health classes. Also, participants who received the “Active Guide” brochure and related documents outlining the benefits of reducing sedentary behavior were 16 (53.3%) in Group A and 19 (47.5%) in Group B.

2.3.6. Hypothesis 2 methods

In previous study [19], we showed that intervention may reduce sedentary behavior in elderly people one year later. However, additional studies were required to assess whether the decline in sedentary behavior persisted after two years. The second phase study involved a similar follow-up survey and was conducted between 29 April and 31 May 2018.

2.3.7. Previous study methods

The RCT study consisted of two groups of elderly people living in Japan. Our RCT study, however, instead of random sampling, random assignment was used. Shortly after receiving their baseline results, the intervention group also received the “Active Guide” brochure and related documents outlining the benefits of reducing sedentary behavior. Members of the control group received only the baseline test results.

The sample size was calculated using significance level, power, difference between two groups, and standard deviation (SD) by paired t-test. To have a sample exceeding the calculated sample size of 39 per group is good and could be explained in those terms rather than as deliberate sampling choice to have 10% attrition allowance.

  1. L.58-60.

(I have inserted a purpose at the end of the introduction.)

we clarified how the frequency of participation in a health class affect reduction in sedentary behavior after two years and whether decreases in sedentary behavior in elderly who participated in a health class persist two years after the end of intervention.

3.L.71-72.

(I added “initial eligibility criteria”.

Twenty-two participants were excluded because they did not participate in more than 75% of the 12 sessions during the trial period.

Point 2.3.6 the authors cannot refer to the results of a study in the materials and methods section.

(Answer)

(I rewrote “2.3.6” as follows.)

2.3.6. Hypothesis 2 methods

In previous study [19], we showed that intervention may reduce sedentary behavior in elderly people one year later. However, additional studies were required to assess whether the decline in sedentary behavior persisted after two years. The second phase study involved a similar follow-up survey and was conducted between 29 April and 31 May 2018.

(I rewrote “3.2” as follows.)

3.2. Hypothesis 2 results

Breakdown of 71 participants same as Hypothesis 1 were the intervention (number of participants, 35: average age, 73.5 ± 5.7 years) and control group (number of participants, 36: average age, 74.3 ± 5.3 years) (Figure 1, Hypothesis 2). No difference was found between groups in sedentary behavior rate (intervention group, 50.7%/day, control group. 50.6%/day; p = 0.872) at the two-year follow-up (Table 3) despite observing differences after one year (Table 4). On the contrary, the parameters before and after the two-year follow-up were significantly lower in the control group than in the intervention group: ≤ 1.5METs (%/day) (-2.1 vs. -7.0; p = 0.034), 1.6~2.9 METs (%/day) (6.4 vs. 10.1; p = 0.041), 3~5.9 METs (%/day) (-1.5 vs. 2.1; p = 0.044) (Table 3).

Point 2.3.8 The reviewer suggests that the authors describe the statistical analysis in more detail.

(I rewrote “statistical analysis” as follows.)

2.3.8. Statistical analyses

Data analysis was conducted by a researcher blinded to the groupings of the participants. Continuous variables were presented as the mean ± SD. Welch’s t-test was used to compare the averages of continuous variables. If there were variables present that could affect the results, they were adjusted using multiple regression analysis by these variables prior to intervention. The threshold for significance was p < 0.05. All calculations were performed using STATA, version 14 (STATA Corp LLC., 4905 Lake way College Station, Texas 77845-4512 USA).

Results section. the reviewer advises the authors to transfer figure 2 to the methods section. Likewise, the reviewer advises the authors to write the results so that the found results are more clearly expressed. the use of tables should reduce the text to punts of greater emphasis.

(Answer)

(I swapped “Materials and Method, Result, Tables” to make the results clearer.)

Discussion section: The reviewer suggests that the authors initially highlight the main findings linked to the main objectives and develop them in the discussion, without the need to present them following the chronological sequence of the studies carried out. Conclusions Again the reviewer suggests to the authors to clarify what the contributions of the study are.

(Answer)

  1. L.214-219.

(I rewrote “Discussion” as follows.)

  1. Discussion

There was a statistically significant difference in the rate of reduction in physical activity (≤ 1.5METs) between the group that took the health class and the group that did not. On the other hand, there was no significant difference between the intervention group and the control group. The “Active Guide” brochure and additional information handouts were effective one year after the intervention, they might not be valid after two years in community-dwelling Japanese elderly.

  1. L.268-270.

((I rewrote “Conclusion” as follows.)

  1. Conclusions

In conclusion, (1) continuous participation in health clubs may help reduce sedentary behavior. (2) After two years, the use of the “Active Guide” brochure and related documents may not reduce the sedentary behavior in elderly people living in Japan despite observing a reduction after one year.

Reviewer 2 Report

Overall an interesting study but with some confusion factors. Mostly this occurs where an attempt has been made to marry several studies together and where one has already been published.

I would recommend less use of cut and paste with changes to a few words and instead more rewriting to enable a more productive integration of material from the different sources. It would enable progression to be demonstrated rather than a series of separate studies.

Statistical results are given as p values only, so it is not clear where you have used a Welch corrected t-tests and where you have used a test of medians as stated in line 104. No medians were presented.

Study three is hard to interpret because there is no way of separating out the Active Guide and attendance at health classes. Only part of the sample were given the Active Guide in study one, but it appears that some of those people were in the under 75% attendance group (B) and others were in the 75% or more group (A) for the comparison in study three. We do not know whether one group had more people in it that received the brochure, so it is hard to separate out the class effect from the brochure effect.

Differences seem to be calculated from 2016-2017 and from 2017-2018.  For the title of the study it would be interesting to know whether at the end of the two years there is a decrease from the start point, ie 2016-2018.

Specific suggestions

Page

Concern/suggestion

16-17

Clarify whether differences year on year or from baseline measures

19

Why 75% as the cut-off for participation in classes? Not explained in abstract or later.

61-64

To begin with I questioned how the studies were compatible then a read of reference 19 made me realise that the sample is the same people.  It would be helpful to say that here to reduce concern about linking the sets of information.

It would be more logical to reverse the order of the hypotheses.  Participation in health classes occurred before assessment of the sustained effects of the intervention.

65

Use of the word ‘participant’ rather than ‘subject’ is considered more ethical, certainly in the Western world. It is worth consideration as you amend the paper.

85-91

In this paragraph there is a good level of detail, but no information about how long elderly people attended the once a month classes.  You give 27 topics and say that remaining lectures were repeats – but how many were there? How long would participant attend for?  I assume that more than one topic was included in each monthly lecture or there is over two years of lecture before the repeat information. 12 sessions mentioned in line 125 – so how do the topics and repeat sessions fit together? Some clarity around this structure is needed to prevent readers making false assumptions.

103-104

“Mean physical activity did not show normal distribution”. Good rigour in demonstrating checks and sample a bit small for CLT.  However, use of medians is mentioned but no medians are reported in the results and no reference made to a test of medians to replace the t-tests used (some paired and some independent? But that is not stated).

119

The RCT is random allocation not a randomised control trial in the traditional sense.  Better to say so here and then revisit as a limitation later (as you have done).

125

You say here that there were 12 monthly sessions so over the course of a year.  The reader needs that information earlier to make sense of the paragraph starting at line 85

137

Average sitting time was higher than that identified by literature in line 28 from a paper published in 2011. Are there any more up to date figures from published literature before your own  ‘Active Guide’ study in 2019 using these same participants

127-142

This information is mostly word for word the same as the 2019 paper above.  The last sentence is confusing though. The 86 participants were because that is what came through from the previous study – as illustrated in figure 1.  To have a sample exceeding the calculated sample size of 39 per group is good and could be explained in those terms rather than as a deliberate sampling choice to have 10% attrition allowance.  To say so explicitly would give a less confusing end to the paragraph.

158-159

It would be helpful if the variables used for correction or covariate analysis were identified specifically. How were adjustments made?

169-182

These results are a repeat of the published paper of 2019. The purpose of repetition is not clear. Some kind of integration of results from the studies to draw overall conclusions either here, in the discussion, or both would be really helpful in demonstrating the value of the findings in this paper.

192

The values should be -2.0 and -7.1 given the values presented in the tables from lines 203-208.

198

More impact from showing a significant increase or decrease rather that just a difference. How was the decrease calculated as the group is not the same one as the control or intervention group in the earlier studies.

274-276

The explanation of power is confusing. Do you mean that to achieve power of 80%, 41 people are required per group (more than you had)? Otherwise I do not see why 41 is important as you had less than 40 per group. If you did a post hoc calculation on you sample size with the effect sizes reached by your three analyses (0.614,0.190, and 0.469) you would have reached power of 71.2%, 12.2% and 48.3% for your three percentage change in METs levels.

277

This is the first time ‘Nudge’ has been mentioned – there is no explanation so it is not clear how the conclusion was reached from the results presented nor how it relate to the title.

326

Reference 13.  It is not yet May 2020 so make sure that the date of access relates to the date this source is finally accessed prior to final submission.

Author Response

Reviewer 2

Overall an interesting study but with some confusion factors. Mostly this occurs where an attempt has been made to marry several studies together and where one has already been published.

I would recommend less use of cut and paste with changes to a few words and instead more rewriting to enable a more productive integration of material from the different sources. It would enable progression to be demonstrated rather than a series of separate studies.

(Answer)

I changed to make my manuscript clearer.

Statistical results are given as p values only, so it is not clear where you have used a Welch corrected t-tests and where you have used a test of medians as stated in line 104. No medians were presented.

(Answer)

L.114-116.

(I added these sentences.)

physical activity of each participant used the median of 7 days. We adopted the median of each participant to calculate the average in each group.

Study three is hard to interpret because there is no way of separating out the Active Guide and attendance at health classes. Only part of the sample were given the Active Guide in study one, but it appears that some of those people were in the under 75% attendance group (B) and others were in the 75% or more group (A) for the comparison in study three. We do not know whether one group had more people in it that received the brochure, so it is hard to separate out the class effect from the brochure effect.

(Answer)

L.135-139.

(I added these sentences.)

We adopted a similar classification criterion in Hypothesis because the initial eligibility criterion was a participation rate of over 75% in health classes. Also, participants who received the “Active Guide” brochure and related documents outlining the benefits of reducing sedentary behavior were 14 (46.7%) in Group A and 21 (51.2%) in Group B.

Differences seem to be calculated from 2016-2017 and from 2017-2018. For the title of the study it would be interesting to know whether at the end of the two years there is a decrease from the start point, ie 2016-2018.

(Answer)

Thank you for your advice.

I think that before-and-after comparison is statistically difficult, so I did not do it in this study.

The result of the intervention may be “regression toward the mean”.

Specific suggestions

Page

Concern/suggestion

16-17

Clarify whether differences year on year or from baseline measures

(Answer)

L.17-20.

(I rewrote it as follows.)

Results: There was a significant difference from the previous year in sedentary behavior rate of reduction between groups with health class participation rate of greater than 75% and less than 75%. Furthermore, no difference was found between groups in sedentary behavior rate at a two-year follow-up compared to the previous year despite observing differences after one year.

19

Why 75% as the cut-off for participation in classes? Not explained in abstract or later.

(Answer)

L.135-137.

(I added it as follows.)

We adopted a similar classification criterion in Hypothesis because the initial eligibility criterion was a participation rate of over 75% in health classes.

61-64

To begin with I questioned how the studies were compatible then a read of reference 19 made me realise that the sample is the same people. It would be helpful to say that here to reduce concern about linking the sets of information.

(Answer)

L.64-65.

(I added it as follows.)

The participants in this study were the same as in the previous study.

It would be more logical to reverse the order of the hypotheses. Participation in health classes occurred before assessment of the sustained effects of the intervention.

(Answer)

I changed the position of the hypothesis.

65

Use of the word ‘participant’ rather than ‘subject’ is considered more ethical, certainly in the Western world. It is worth consideration as you amend the paper.

(Answer)

I changed the subject to a participant.

85-91

In this paragraph there is a good level of detail, but no information about how long elderly people attended the once a month classes. You give 27 topics and say that remaining lectures were repeats – but how many were there? How long would participant attend for? I assume that more than one topic was included in each monthly lecture or there is over two years of lecture before the repeat information. 12 sessions mentioned in line 125 – so how do the topics and repeat sessions fit together? Some clarity around this structure is needed to prevent readers making false assumptions.

(Answer)

(I rewrote it as follows.)

2.3.2. Health class

In a previous study (2016-2017), 86 of 90 collaborators participated in more than 75% of 12 sessions during the trial period. In the 2017-2018 study, the number of data obtained was reduced to 71. The breakdown of 71 people was 30 (more than 75% attended health class) and 41 (less than 75% attended health class). Health class sessions were held once a month for 90 minutes, and mainly consisted of lectures (45 minutes) and exercises (45 minutes) for health promotion. During 2016-2017, the average number of participants per event was 50.2±10.3 (minimum, 29; maximum, 62). During 2017-2018, the average number of participants per event was 36.5±8.7 (minimum, 20; maximum, 50). Participants in a health class are to attend each month for two years in principle, but this is not mandatory. The number of registrants was 121 at the end of 2018. The lectures encompassed 27 topics: 14 focused on muscle, bone, and joints, 6 on medical diseases, 7 on health care during 2016-2017. And the same lectures were repeated during 2017-2018. Lectures and exercises were facilitated by doctors, physical therapists, occupational therapists, nurses, acupuncturists, judo therapists, and athletic trainers.

103-104

“Mean physical activity did not show normal distribution”. Good rigour in demonstrating checks and sample a bit small for CLT. However, use of medians is mentioned but no medians are reported in the results and no reference made to a test of medians to replace the t-tests used (some paired and some independent? But that is not stated).

(Answer)

L.114-116.

(I added these sentences.)

physical activity of each participant used the median of 7 days. We adopted the median of each participant to calculate the average in each group.

119

The RCT is random allocation not a randomised control trial in the traditional sense. Better to say so here and then revisit as a limitation later (as you have done).

(Answer)

L.146-147.

(I added these sentences.)

Our RCT study, however, instead of random sampling, random assignment was used.

125

You say here that there were 12 monthly sessions so over the course of a year. The reader needs that information earlier to make sense of the paragraph starting at line 85

85-91

(Answer)

(I rewrote it as follows.)

2.3.2. Health class

In previous study (2016-2017), the data from participants of both groups who participated in over 75% of the 12 sessions during the trial period were used for the final analysis. Of the 90 participants, the data from 86 were included. In 2017-2018, the participants were divided into two groups: those who attended 75% or more of the health classes and those who attended less than 75%. Health club sessions were held once a month for 90 minutes, and mainly consisted of lectures (45 minutes) and exercises (45 minutes) for health promotion. During 2016-2017, the average number of participants per event was 50.2±10.3 (minimum, 29; maximum, 62). During 2017-2018, the average number of participants per event was 36.5±8.7 (minimum, 20; maximum, 50). Participants in health class are to attend each month for two years in principle, but this is not mandatory. The number of registrants was 121 at the end of 2018. The lectures encompassed 27 topics: 14 focused on muscle, bone, and joints, 6 on medical diseases, 7 on health care during 2016-2017. And the same lectures were repeated during 2017-2018. Lectures and exercises were facilitated by doctors, physical therapists, occupational therapists, nurses, acupuncturists, judo therapists, and athletic trainers.

137

Average sitting time was higher than that identified by literature in line 28 from a paper published in 2011. Are there any more up to date figures from published literature before your own ‘Active Guide’ study in 2019 using these same participants

(Answer)

“Reference1” was a self-reported value and “Reference27” was an active meter value.

L.30-31.

(I added these sentences.)

These values are based on participants' self-reports and are less accurate but general trends can be seen.

127-142

This information is mostly word for word the same as the 2019 paper above. The last sentence is confusing though. The 86 participants were because that is what came through from the previous study – as illustrated in figure 1. To have a sample exceeding the calculated sample size of 39 per group is good and could be explained in those terms rather than as a deliberate sampling choice to have 10% attrition allowance. To say so explicitly would give a less confusing end to the paragraph.

(Answer)

I rewrote these sentences.

L.152-154.

To have a sample exceeding the calculated sample size of 39 per group is good and could be explained in those terms rather than as deliberate sampling choice to have 10% attrition allowance.

158-159

It would be helpful if the variables used for correction or covariate analysis were identified specifically. How were adjustments made?

(Answer)

L.158-159.

(I added these sentences.)

they were adjusted using multiple regression analysis by these variables prior to intervention.

169-182

These results are a repeat of the published paper of 2019. The purpose of repetition is not clear. Some kind of integration of results from the studies to draw overall conclusions either here, in the discussion, or both would be really helpful in demonstrating the value of the findings in this paper.

(Answer)

I rewrote these sentences.

192

The values should be -2.0 and -7.1 given the values presented in the tables from lines 203-208.

(Answer)

I think it's -2.1 and -7.0 because it's the value in Table 3.

198

More impact from showing a significant increase or decrease rather that just a difference. How was the decrease calculated as the group is not the same one as the control or intervention group in the earlier studies.

(Answer)

L.230-232.

(I added these sentences.)

The reduction in sedentary behavior (≤ 1.5 METs) was as large as -6.7% for Group A and -2.4% for Group B. This may be due to the fact that there were 9 dropouts in 2018 compared to the previous year, but the average percentage of those 9 who resigned in 2017 was 64.7% (≤ 1.5 METs).

274-276

The explanation of power is confusing. Do you mean that to achieve power of 80%, 41 people are required per group (more than you had)? Otherwise I do not see why 41 is important as you had less than 40 per group. If you did a post hoc calculation on you sample size with the effect sizes reached by your three analyses (0.614,0.190, and 0.469) you would have reached power of 71.2%, 12.2% and 48.3% for your three percentage change in METs levels.

(Answer)

I have deleted these sentences.

277

This is the first time ‘Nudge’ has been mentioned – there is no explanation so it is not clear how the conclusion was reached from the results presented nor how it relate to the title.

(Answer)

I have deleted these sentences.

326

Reference 13. It is not yet May 2020 so make sure that the date of access relates to the date this source is finally accessed prior to final submission.

(Answer)

I corrected the date from May 1 to March 1.

Round 2

Reviewer 1 Report

The authors have made substantial changes to the manuscript, greatly improving its quality and the reader's understanding.